# Multi-Dimensional Accessibility Barriers in Care Services for the Rural Elderly with Disabilities: A Qualitative Study in China

**DOI:** 10.3390/ijerph18126373

**Published:** 2021-06-12

**Authors:** Yuan Wang, Caiyun Qi

**Affiliations:** Department of Labor and Social Security, Jilin University, Changchun 130012, China; wywy19840502@jlu.edu.cn

**Keywords:** rural elderly with disabilities, care services, accessibility barriers, welfare pluralism

## Abstract

This research covers a multi-dimensional investigation into accessibility barriers in care services for older people with disabilities in rural China. In-depth interviews with 13 rural disabled older people in China were conducted using qualitative methods. Based on a welfare pluralism approach, the results showed that in comparison with urban areas, care services for disabled older populations in rural areas are more subject to social barriers. This can be seen in the limited state (lack of resources, rigorous eligibility qualifications, uneven distribution, and irregular implementation); the absent market (low levels of consumption, high cost pressures, self-exclusion, and traditional cultural constraints); absent NGOs and volunteers (difficulties in access for NGOs and volunteers outside the area and formation difficulties of local NGOs and volunteers); as well as low-quality care in households and communities (unprofessional care from the spouse, unsustainable care from children, and unavailable community-based care). A multi-subject support network should be established to remove accessibility barriers to care services for older people with disabilities in rural areas through active intervention and interaction. The results of the research provide insights that will aid in the formulation of future social care service plans and health policies for rural older people with disabilities.

## 1. Introduction

One of the most important and visible social problems today is the increase in the aging global population. As life expectancy increases, the number of older people with disabilities at risk of chronic illness or injury also inevitably increases, bringing significant societal challenges. Physical dysfunction and chronic illness reduce the quality of life of vulnerable older populations, leading to extensive demand for medical and care services [1]. Therefore, improving care conditions for disabled older people has become an important topic in academia.

In China, data from the Second China Sample Survey of Disabled Persons (2006) showed that 44.16 million people with disabilities were aged over 60, accounting for 53.24% of all disabled people [2]. As the aging of the population increases, the scale and growth rate of the disabled older population in China in the future will be significant. The prevalence of disabled people over 60 is predicted to increase by more than 7 million every 5 years, reaching 103 million by 2050, 2.3 times the number in 2006 [3]. More importantly, 70.42 % of older persons with disabilities live in rural areas, totaling about 31.1 million in 2006 [2]. These older people have multiple social identities such as countryfolk, the disabled, and the aged, and the most vulnerable face serious economic pressure as well as health care needs [4,5,6,7]. However, their special care needs and barriers to access have been somewhat neglected in existing research, as previous studies concerning the disabled elderly have usually been focused on urban subjects (largely due to investigation convenience) and there is scarce relevant extant research on the accessibility barriers faced by the disabled elderly in rural areas.

Most studies concentrate on describing the phenomenon of accessibility barriers—the high but unmet demands for social services by rural older people with disabilities. These demands include a customary and safe environment, adequate income for living, support from family members and friends, and access to health care services [8]. However, satisfying these demands is significantly more difficult for the disabled elderly in rural China than for their urban counterparts, particularly in health care services. Research has highlighted poor living conditions and self-care ability, the risk of disease, and the severe aging trend as well as poor social interaction and poor care services for disabled older people in rural areas [9,10]. Dependence on family care is high, with low rates of institutional welfare support and weak informal support from society [11,12]. The Fourth Sample Survey of Living Conditions of the Elderly in Urban and Rural China (2016) found that 89.43% of the rural disabled elderly requiring care were receiving it, but the major providers of the care were mostly family members (93%), among whom nearly half (43.48%) were spouses, 28.64% sons, 10.08% daughters-in-law, and 10.35% daughters. Less than 1% of people received professional care from nursing institutions [13]. Family care cannot meet diverse health needs due to limited professional levels, fewer services, and lower service frequency [14,15,16]. In such situations, “muddling along” generally describes the mentality of rural older people with disabilities [17]. In addition, their problems are compounded by the social issues of aging, family miniaturization, and rural hollowing-out, producing families with multiple disadvantaged characteristics such as multiple disabled family members, two-generation disabled elderly families, empty nest families with disabled older people, disabled elderly families with the loss of a child, and families with both elderly and disabled people, thus leading to increasingly complex barriers for older people with disabilities to obtain care services [6,18].

A small group of scholars have tried to analyze the causes of accessibility barriers, mainly from three interpretative perspectives. The geographical location approach sees cost and traffic as the major reasons, as health infrastructure and professional care services are concentrated in cities far away from rural areas [10,19]. The social capital approach looks at the lack of social support networks in rural areas to share and exchange information on public services [20]. Some scholars have focused on the cultural perspective, seeing the conflict between Chinese traditional culture and the rapid expansion of the market as producing negative consequences in family care [21].

It is crucial to study multi-dimensional accessibility barriers to care services for the rural elderly with disabilities, as these barriers are closely related to the improvement of their quality of life. The existing literature has shown that, in rural China, the elderly with disabilities face severe obstacles to obtaining care services. However, their special needs are significantly neglected in both policy practice and academic research. Additionally, most of the relevant reviews describe the phenomenon in a fragmentary way, without further exploration of the reasons. Moreover, they simply summarize the institutionalized care services provided by the state and the non-institutionalized care services provided by the traditional family, an approach that ignores the multiple subjects of the service supply and leads to a narrow understanding of the care supply-demand imbalance.

To better understand the multi-dimensional factors that create the barriers, this paper focuses on the service providers, analyzing the issue from the perspective of welfare pluralism. After the welfare crisis of the 1970s, some scholars realized that the sources of social welfare should be diversified. Rose first put forward the theory of a welfare mix, identifying that the supply of social welfare comes mainly from three sectors—the state, the market, and households [22]. Johnson joined the volunteer sector on the basis of Rose’s theory, identifying four sectors of welfare provision, thus forming an overall framework of welfare provision [23]. The concept of welfare pluralism has received widespread attention in the field of social policy and has been used as a guideline in the formulation and implementation of old-age service policy and long-term care insurance policy in China [24,25]. To explore the reasons behind accessibility barriers, this paper adopts the useful perspective of the welfare pluralist and gives a comprehensive framework to capture various aspects of institutional, social, cultural, and other macro factors behind accessibility barriers, avoiding individual analysis of rural older people with disabilities.

Using a qualitative method, this research interviewed 13 rural elderly with disabilities about their experiences and feelings in mainland China. By analyzing these stories through the perspective of the welfare pluralist, we found diverse obstacles coming from the state, market, NGOs, volunteers, households, and the community.

## 2. Materials and Methods

### 2.1. Design

Studies into aging and disability should provide a rich empirical examination of how older people with disabilities experience barriers and impairments. At present, much of the research on the rural elderly with disabilities has been based on the quantitative method; however, a fixed questionnaire can often ignore the indelible individual experience of the interviewees, whereas a qualitative method can provide additional insight into this issue. To obtain more vivid and richer knowledge, we adopt an inductive approach in this paper, using a qualitative method to search the subjective care service experiences of the elderly with disabilities in rural China through the collection of in-depth interviews. An in-depth interview is a type of unstructured, direct, deep, and one-to-one interview, which is an appropriate way to collect data on the potential motives, experiences, attitudes, and emotions of the respondents regarding a certain issue [26]. In this study, we highlighted the personal stories of rural elderly with disabilities, their personal experiences, and their reflections on interactions with diverse service providers. The analysis of behavioral characteristics and welfare experience in specific fields can answer the question of how policy providers can cause severe accessibility barriers in care services. The study was conducted in Jinan, the capital of Shandong Province, an eastern coastal city in China. In 2020, there were 154,902 citizens with registered disabilities in Jinan, of which 74,956 were elderly with disabilities, accounting for 48.4% of the total. Among the elderly with disabilities, 56,088 were registered as rural residents, accounting for 74.83% of the disabled elderly. Because of physical, intellectual, and disease-related obstacles, most were facing severe demand for care services.

### 2.2. Participants

We recruited participants through the local village committees, an administrative institution in rural China responsible for the provision of social welfare and social services for the rural disabled and rural elderly, including cash allowances, health rehabilitation, care services, and medical reimbursement support. With the help of village committees, we recruited participants through two villages—a suburban village (Village A in the paper) affected by China’s economic development and a more traditional village (Village B in the paper). The subjects of the study were 13 rural older people with disabilities from non-natural aging factors, such as congenital disability, disease, or accident. Of these, 7 resided in village A and 6 were in village B. The participants were recruited through purposive sampling, considering specific criteria related to the research objectives [27]. The criteria were as follows: (1) age of 60 or above (the standard for the elderly classification is 60 years old and above in China); (2) a disability license (an official document for people with disabilities) with a duration of no less than a year; (3) registration as a rural citizen (known as “hukou” in China) who has lived in the countryside for a long time; (4) clear cognition of the participant and no communication barrier with the researchers; and (5) informed consent and voluntary participation. At the same time, we fully considered diversity in age, gender, and type and degree of disability. We agreed with Crabtree and Miller that it is better to spend more time, more attention, and talk with fewer people than engaging more people in conversations that lack depth [28]. Therefore, we preferred to obtain enough data from a smaller number of interviewees. When much of the data was repetitive, we felt it had reached a saturated state.

The respondents were aged between 60 and 78 years, and included 6 males and 7 females. Of these, 4 participants had a severe disabilities and 9 were slightly disabled. Eleven were empty-nesters, and two were older people living alone. Eight people were disabled due to disease (sequelae caused by diseases such as diabetes, hypertension, cerebral thrombosis, heart disease, rheumatism, and arthritis), three were born disabled, and two were disabled by an accident. Five were from multi-disabled families (a multi-disabled family tends to have more complex care needs and more abundant personal experiences). Table 1 describes the demographic data, nature of disabilities, and primary caregivers of the participants in the study.

### 2.3. Data Collection

The interviews took place in January and February 2019. The second author completed all one-on-one in-depth interviews in Mandarin Chinese or in dialect, each ranging from 30 min to 1 h. Before the interviews, we contacted the participants and provided general information related to our research, including the purpose, subjects, process, duration of the study, and the research schedule. After obtaining the consent of the interviewees, the interviewers recorded the dialogues through on-site audio-recording and note-taking to collect raw data. The notes included the core contents of the interview, the physical and mental state of the interviewee, as well as behaviors and attitudes observed in the interview. All interviews took place in the participants’ homes.

The first author and second author were researchers who had received training in qualitative methods and interview techniques. They designed the semi-structured interview schedule based on the purpose of the research. Interview questions looked at participants’ experiences of obtaining care services from the state, market, non-governmental organizations (NGOs), volunteers, households, and the community. Table 2 details the content of the interview schedule.

### 2.4. Data Analysis

After the interview, we converted the audio-recordings into Chinese, word by word, as well as the interview diary and notes, and selectively translated some information into English as required to present the findings. This research used a qualitative thematic analysis method to analyze the data [29]. Thematic analysis is a popularly implemented method of qualitative research. By applying this method, researchers can gain better and deeper understanding of the participant’s attitude, vision, feelings, and thought reflections on care services from a dataset that has been collected from rural elderly with disabilities, and extract the core theme [30].The software package NVivo 10.0 (NVIVO, Göteborg, Sweden) was used for analysis, and the raw data were composed of the participants’ narratives that we coded over 3 stages: We read the raw data repeatedly to familiarize ourselves with all the dimensions of the data, and then extracted meaningful statements to generate initial codes (e.g., “insufficient support resource investment”, “strict criteria”, etc.). Following the initial coding, we focused on the broader level analysis by collating all codes into themes (e.g., “limited state”, “absent market”, and so on). We then reviewed the themes to test whether they related to the initial codes. To ensure the reliability of the analysis, we returned the initial codes and theme codes to the respondents and invited suggestions for these codes.

### 2.5. Ethical Considerations

During the in-depth interview process, our research strictly followed the procedures of informed consent, non-harm, and confidentiality of participants. Before the interview, we explained the purpose and use of the interview as well as the recording requirements in detail, and all participants had signed the informed consent. Second, due to the relatively fragile physical and mental condition of rural older people with disabilities, as well as a tendency towards self-deprecation, the interviews were carried out after full discussion with the workers of village committees to protect the interviewees during the interview process. Third, for confidentiality, all identifying information was anonymous during transcription and translation. Any sensitive material from the interviews was technically processed.

## 3. Results

Using the perspective of welfare pluralism, we divided the factors of accessibility barriers to care services into four themes: the limited state, absent markets, absent NGOs and volunteers, and low-quality household and community care.

### 3.1. Limited State

The limited state refers to the severe restriction in the provision of various health services and care services, as compared with advanced welfare countries. The main barriers are insufficient resource investment, overly strict eligibility examinations, uneven distribution, and irregular implementation.

Support resources from the state and government are seriously insufficient and not sustainable. The main form of support lies in short-term subsidies and donations. Of the social services related to medical health and care, only a serious disease pension is provided, with no more than half of the cost reimbursed—too little to solve the most severe problems of the rural disabled elderly and their families.


*The government says that work for the disabled includes medical rehabilitation, education, employment, long-term care, social security, and so on. But, in fact, children’s rehabilitation, education, and employment are still the key work. Old people like me only have two low-level subsidies and a serious disease pension. But I am getting older and older, and we cannot recover. How am I going to live? *
*(P-2)*

While choosing welfare candidates eligible for a new policy of care service, the method of “doing subtraction” is often adopted to reduce the number of service users. For example, for home-based care the procedures are as follows: in addition to the regulated poverty line standard, the applicant is excluded if they have a son, a big house, a car, or if they use expensive household appliances of the same type. Such strict exclusions render only two to four rural disabled old people in a village eligible to qualify, and there are many disabled older people with urgent care needs who are ineligible. As the participants have demonstrated:


*Every year I apply for care assistance, but never succeed, because I have two sons. But what’s the use of two sons? They have debts of more than a million, and everything in my house is taken away to pay off the debts. They can hardly live themselves; how can they care for me? *
*(P-9)*


*[The village committee] says that I have sons, so I can’t apply. But my eldest son is also disabled. It is me who takes care of him, not he who takes care of me. *
*(P-13)*

Furthermore, there is a huge gap between urban and rural disabled old people in the development of social service support. In urban areas, the elderly care service framework has been established with the “household as the core, community as the dependence, and professional institutions as the supplement” [29]. Old people with disabilities can enjoy day care centers, rehabilitation centers, dining halls for the elderly, door-to-door nursing, and other professional services, whereas in rural areas there is nothing. In practical implementation, affected by the level of economic development and policy radiation scope, day care centers and rehabilitation centers are merely a door plate and are virtually useless. Dining halls for the elderly and door-to-door nursing are still in discussion but have not been implemented. The welfare differences in the two areas are widening.


*We don’t have a day care center. The rehabilitation center is in the sub-district office, which is too far to go. I have never gone there. The rural areas are different from the urban. There are too many disabled old people in the countryside, and the government can’t take care of all of them. We have to depend on ourselves and our family, not the government. *
*(P-4)*

This disparity is further increased by the irregular implementation of street-level bureaucrats. Our investigation found that some street-level officers take social welfare for the disabled elderly as a way to obtain favors and realize personal interests to expand their relationship capital. That is, they give the limited social benefits to their family members, relatives, and friends. In turn, people in real need are unable to get the support they deserve, further squeezing the opportunities of those in need to obtain health welfare and care services.


*I don’t believe in the government… I have never heard of this kind of care service. If there is such a service, it must have been secretly given to their own relatives. *
*(P-13)*

### 3.2. Absent Market

The term absent market refers to the fact that the market does not provide care services for rural elderly persons with disabilities. Although with the rapid development of the Chinese market economy, the market plays an increasingly prominent role in the socialized care for disabled older population, this is true in urban areas only. In rural areas, there is no professional provision of care for senior citizens. The reasons are as follows:

First are the economic limitations. Rural households tend to have low income and poor living standards. It is likely that there would be no potential customers for old-age care services charged at cost in rural areas.


*There is not a charged care service, we could not pay for the care even if there were. Eating expenses are a problem for us, who would be willing to spend money on care? *
*(P-10)*

Second are geographic restrictions. Rural areas are often remote with poor roads, resulting in higher costs in providing care services than in urban areas. This does not conform to the low-cost principles of the market.


*Hiring a caregiver? That’s for the city, in the village, no one pays for that. No caregiver wants to come this far. *
*(P-12)*

Third are barriers of exclusion. On one hand, exclusion may result from the market for elderly service provision, as the market prefers customers with slight impairment rather than higher degrees of disability. On the other hand, it may result from the self-exclusion of disabled older people in rural areas. In Chinese society, referring to someone as disabled elderly often suggests the person is unable to do anything for themselves, and has negative connotations. This notion further influences negative attitudes toward care. In most cases, the root cause is self-exclusion, but it is often reflected through the market.


*In a care service, people would dislike me because I am dirty, messy, and poor. My own children dislike me when caring for me, not to speak of outsiders. *
*(P-11)*

Finally, there are obstacles of traditional ideas. In rural areas, the traditional idea of “raising children for old age” still exists with the belief that the responsibility of care of the elderly should be shouldered by their children. Consequently, paying for care services not only makes rural disabled elderly lose face, but also gives their children a bad reputation.


*Those who have no sons or daughters are taken care of by others, and arrangements are made to go to a nursing home. Otherwise, people will laugh at you. *
*(P-5)*

### 3.3. Absent NGOs and Volunteers

As with the absent market, NGOs and volunteers are not found in rural areas. Care services are faced with the double barrier of difficulty of access by outside NGOs and volunteers, as well as difficulties in forming local NGOs and voluntary groups.

It is difficult to encourage NGOs and voluntary groups to work in rural areas. NGOs tend to be located in cities and volunteers are usually college students, mainly providing services in urban areas. The countryside is often neglected due to its remote geographical location and loose housing distribution. During the interviews, it transpired that many elderly people with disabilities in rural areas had not even heard the words “volunteer”, “nongovernmental organizations”, or “free service”. Most participants said that they had not been exposed to any welfare services offered by NGOs, nor had they received any help from volunteer teams.


*(Interviewer: Have you had any volunteers or anything?) What is a volunteer? (Interviewer: It’s a person who helps you without being paid for it.) I don’t look for them. I dare not trouble them. *
*(P-3)*

It is also hard to form local NGOs and volunteers. China has been promoting models of mutual support for the aged such as “time banking” and “Yizhuang” (a charitable organization popular in rural China) in recent years, but with little success. There are two main reasons for this:

The number of people able to provide care services in rural areas has dropped sharply due to urbanization. In today’s China, the young and middle-aged labor force tend to work in the city, and many old people are needed in the cities to bring up their grandchildren, known as “skip-generation raising”. Consequently, those left behind in the rural areas are mainly old, weak, sick, or disabled, which is challenging to the integration of voluntary groups.


*If you are in good health, you go to work for others. If not, you stay at home to do farm work and cook. Everyone is too busy. Who is willing to help an old woman like me? *
*(P-8)*

The other reason is that rural older people with disabilities are often alienated as people without reciprocal benefits. The model of rural mutual support for the aged relies on the informal relationship maintained by the primary group, mostly based on personal favors. In essence, the Chinese “favor” has an egoistic motivation. Only when it is “good” or provides reciprocation for oneself and one’s family can the relationship survive. Hence, there is a contradiction in this type of informal support. The poorer the individual and the higher the degree of disability (those with the most urgent care needs), the less support they receive from relatives, friends, and neighbors.


*I have no friends. I have no money to buy anything for others. I will not go to others’ homes and they will not come to mine. *
*(P-3)*

### 3.4. Low-Quality Household and Community Care

Family care is the main form of care for the disabled elderly in rural areas, mostly through care from the spouse, supplemented by care from their children. The model of community-based service is far from being established. Such care service is characterized by low professionalism levels, being a heavy burden on the caregivers, and is difficult to sustain; thus, we call it “low-quality household and community care”.

Our findings revealed that family members are the major providers of the care service. The support of rural elderly people with disabilities mainly comes from the social network maintained by personal relationships such as spouses, offspring, and relatives and is characterized by informality and discontinuity. This can be divided into two main categories.

The first type is care from the spouse. Rather than the intergenerationally reproduced care model of “raising children for old age”, the current daily care work is mostly done by the spouse. In a family with two disabled people, the less disabled one takes care of the more seriously disabled. Physical limitations result in elderly caregivers only being able to provide simple care services, and often having difficulties in completing actions such as rubbing, bathing, or massage, negatively impacting the health of rural elderly people with disabilities. More importantly, elderly caregivers with low levels of education are often unable to obtain information on caregiving or to buy medicine independently, ultimately affecting the care results.


*In fact, we are both disabled. I’m physically handicapped, she (the wife) is slightly intellectually disabled. She usually takes care of me. She can cook and do simple massage but can’t wipe my body. I have to wait for my daughter to do it and she comes back only on Saturdays or Sundays. *
*(P-10)*


*Ordinarily, my husband take care of me. But when I was in the hospital for a cerebral infection, it was my daughter who helped me. My husband and I can’t read and didn’t know what to do in the hospital. *
*(P-12)*

The second type of care is given by sons and daughters, who usually provide regular care once or twice a week, or once a month. This kind of care is often of short duration and low in frequency, and usually unable to complete periodic rehabilitation and other projects. Additionally, the interviews revealed that with the change in China’s family planning policy, the sons and daughters have a greater task in raising their own children. It is becoming more difficult to balance the responsibility of supporting both the elderly and children. Evidence from the respondents’ narratives uncovers that:


*People of my age have gone to the cities to babysit their grandchildren for their sons. Now my son and my daughter-in-law take care of their child by themselves and have a lot of pressure, so I try not to trouble him too often. *
*(P-8)*

Unfortunately, community-based care is far from being established in rural China. A rural village is a community, with the village committee office at its core. The model of community care service requires the community to be a platform to integrate various service resources and to provide services for the elderly, such as assisting with meals, cleaning and bathing, and medical treatment. However, village committees are highly dependent on state resources and have the primary task of responding to the requirements of those external institutions, with a work style based on routine and formalism. They regard community-based care as an administrative task assigned by the state and local government, with their main goal to complete the task “in form” without necessarily meeting the user’s needs. As front-line service providers, they are often busy writing reports and submitting materials to prepare for inspection by superiors. As mentioned before, day care centers, dining halls for the elderly, and other services tend to nominally fulfill, rather than to truly serve, the needs of the disabled elderly in rural areas.


*No one in the village committee usually comes to see us, but they sometimes inform me to fill in a form. *
*(P-6)*


*Last year, two old people in the village were burned to death. I am afraid. Before this, I heard the village committee say that a “bell for help” would be installed in the homes of the elderly, and then they said that it is not allowed by the leaders. If the bell had been installed, the two old people wouldn’t have died. *
*(P-2)*

## 4. Discussion

Through a survey of two rural areas in China, we found that elderly people with disabilities face multi-dimensional access barriers. This is not an exceptional case. With the rapid aging in China, most of the disabled elderly face these dilemmas. As we noted in the introduction, nearly 90% of disabled elderly people have care needs that are basically met by informal care services provided by family members. Therefore, it is necessary to establish a long-term nursing care system and other aging policies to meet the needs of the rural elderly with regard to those potential influencing factors [31]. Many other countries are also facing these obstacles, such as India, Lebanon, etc., but their specific manifestations may be slightly different [32,33].

To solve the barriers of accessibility in care services for rural elderly people with disabilities, we need to establish a diversified care service provision. This will rely not only on the leading role of the state and the active intervention of the public sector, but also on market mechanisms, NGOs, volunteers, communities, and households, so as to achieve Pareto optimality in the allocation of specific care services.

From the outset, the state should actively undertake the main responsibility of developing the care service system. In addition to formulating relevant policies, the state should increase fiscal spending, expand the scope of supply, and provide more medical rehabilitation and health care services for disabled elderly people in rural areas to address the enormous gap between service supply and real demand. In evaluating the eligibility of service candidates, it is necessary to start from the demand side—to target and respond to rural elderly people with disabilities according to their individual needs. In aiming for better efficiency in the implementation of rural elderly care service policy, the realization of urban–rural equality is still the key method. In fact, rural areas have their own special environments, and the current pursuit of rural–urban consistency in policy leads to substantial inequality. To resolve this, the state and government should innovate the content, form, and supply model of welfare services, and design a unique care service system suitable for disabled elderly people and their families in rural areas. Additionally, strengthening the supervision of policy implementation cannot be ignored, as only by effective supervision can the policy work well.

It is essential to give full play to the function of the market in providing high-quality, personalized, and diversified products in the care service supply process. Firstly, the government should encourage disabled elderly people in rural areas to buy care products, for example, through tax incentives and government subsidies, to expand the local consumption market. Secondly, care service companies could make full use of the rural left-behind labor force of women, establishing an occupational and professional local work team through recruitment and vocational training to solve the problem of service costs. Finally, in the process of accelerating market cultivation, the focus should be to change the social concept of the rural disabled elderly and their family members. We should, through media publicity and policy mobilization, move away from traditional ideas of “raising children for old age” and recognize the market as a natural force, to correctly understand care service products and market transaction behavior.

Following on from this model in rural areas, it would be valuable to have guiding third-party forces such as NGOs and voluntary groups participate in the support network for the supply of care services. The government should enact a variety of legal policy incentives such as tax preferences, private management of public institutions, private institutions with public assistance, and government funding and financial subsidies to encourage urban NGOs and volunteers to get involved in rural services. Furthermore, in the wave of rural revitalization in present-day China, the state has created thousands of jobs in rural areas. We need to take advantage of this opportunity to encourage more young people to stay in rural areas as the human capital needed to vigorously foster and cultivate local NGOs and voluntary groups. It is crucial to transform individual consciousness into public consciousness. Traditional notions around small families also should be broken up, village members should be encouraged to help each other, and a supportive social environment created to assist the elderly and their family members, all of which are important prerequisites for the formation of localized voluntary groups.

Lastly, it is necessary to consider changes in family structures, functions, and social environments to develop family vitality again. Community support networks should be established to help elderly spouse caregivers with the latest policy information and to improve their knowledge and skills. Regular respite care services can also relieve serious care pressure on elderly caregivers. In terms of care from offspring, the state should take the work–family balance as an important guiding principle. The methods of the flexible work system, family responsibility leave, or a compressed work cycle are profitable for employees to reduce the loss of opportunity cost and to mobilize the enthusiasm to provide parental care. In terms of community-based care, it is recommended that an interactive platform in the community be set up involving local government, social organizations, community, volunteers, family caregivers, and rural disabled elderly people for communication, resource integration, and service allocation. The state, NGOs, and the community should work together to build rural community centers in health services, day care, rehabilitation, family services, intelligent care services, and dining areas for the elderly. These centers would provide disabled elderly people and their families with health guidance, management of chronic disease, door-to-door rehabilitation, short-term care for the old, mental care, psychological counseling, and other comprehensive services, thus preventing accessibility barriers and unexpected deaths. In addition, we should strengthen the supervision and regulation of village committees.

This research had three main limitations. As it was the first attempt to explore accessibility barriers in care services for rural elderly people with disabilities, it was difficult to obtain information from existing reviews. Secondly, as the research was based on a small-scale qualitative research method, the applicability of the conclusions to other regions and situations requires further discussion and reflection. Lastly, considering cross-cultural challenges, the results and discussions we have put forward may not adapt well to practices in western countries. Therefore, additional quantitative and comparative research on care services for rural elderly people with disabilities should be conducted in the future.

## 5. Conclusions

This study was based on a welfare pluralism approach to provide a framework for outlining the accessibility barriers in care services for rural elderly people with disabilities in China and the reasons behind them. Based on the analysis of the interview results, the study’s major finding revealed that rural elderly people with disabilities are facing severe multi-dimensional barriers. These arise from the unsuitable actions of multiple service providers, including the state, the market, NGOs, volunteers, households, and the community. Their actions and attitudes have a negative impact on the health of disabled elderly people in rural areas.

Specifically, because of the state’s austerity fiscal expenditure, there are issues such as insufficient resource investment, strict application requirements, uneven resource distribution, and irregular implementation for the provision of care services for rural elderly with disabilities. Under market principles, markets tend to search for more potential customers and reduce service costs. However, elderly with disabilities in rural areas are regarded as “useless” and “fragile” and only their sons and daughters can take care of them in traditional Chinese culture, resulting in an absence of formal services. Moreover, it is difficult for NGOs outside the rural areas to get involved, and the lack of local NGOs and volunteers is the main dilemma. Households are the main provider of care services, but family endowment is inconsistent and unsustainable, causing difficulties for the provision of high-quality care services. Within community-based care, the operation of “formalism” is in evidence, hindering both the accurate implementation of the new model and measures to respond to the needs of rural elderly with disabilities.

## Figures and Tables

**Table 1 ijerph-18-06373-t001:** Description of research participants.

No.	Gender	Age	Disability Type	Disability Degree	Cause of Disability	Multi-Disabled Family	Primary Caregivers
P-1	Female	67	Physical and Visual	3rd degree	Disease	Yes	Spouse
P-2	Female	62	Hearing	4th degree	Accident	No	Daughter
P-3	Female	71	Hearing	4th degree	Disease	Living alone	None
P-4	Female	62	Physical	3rd degree	Congenital	Yes	Spouse
P-5	Male	62	Physical	2nd degree	Disease	No	Spouse
P-6	Male	68	Physical	4th degree	Congenital	Living alone	None
P-7	Female	73	Physical	2nd degree	Disease	Yes	Spouse
P-8	Female	62	Physical	3rd degree	Disease	Yes	Spouse
P-9	Male	66	Physical	1st degree	Disease	No	Spouse
P-10	Male	65	Physical and Speaking	2nd degree	Disease	Yes	Spouse
P-11	Male	61	Visual	3rd degree	Congenital	No	Spouse and daughter
P-12	Female	72	Physical	4th degree	Disease	No	Spouse
P-13	Male	78	Physical	4th degree	Accident	No	Son and daughter

**Table 2 ijerph-18-06373-t002:** Semi-structured interview schedule.

Pre-Established Categories		Questions
Introductory questions		1. How is your health? 2. What kind of care do you need?
Key questions	State	1. What care services do you receive in the country? 2. What barriers do you face in obtaining national care services?
	Market	1. What care services have you purchased from the market? 2. What barriers do you face in obtaining care services from the market?
	NGOs and volunteers	1. What care services do you receive get from NGOs and volunteers? 2. What barriers do you face in obtaining care services from NGOs and volunteers?
	Household and community	1. What care services do you receive from your family and community? 2. What barriers do you face in obtaining care services from your family members? 3. What barriers do you face in obtaining care services from the community or rural residents’ committee?
Ending questions		1. Is there anything missing or that should be added to what we have talked about today?

## Data Availability

The data presented in this study are available on request from the corresponding author. The data are not publicly available due to restrictions of privacy.

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
