# Peer review of "Multi-Dimensional Accessibility Barriers in Care Services for the Rural Elderly with Disabilities: A Qualitative Study in China"

_ijerph, 2021, doi:10.3390/ijerph18126373_

Round 1

Reviewer 1 Report

In my opinion, this study seems to have looked at the problem of care service barriers for elderly with disabilities living in rural areas in terms of welfare pluralism. In order for this study to be more persuasive, I believe that further supplementation of the following will be needed:

1.

In this study, it is believed that the main reason for the lack of care services for the elderly with disabilities in rural areas is the lack of service providers from a welfare pluralism. However, there is a problem in interpreting the lack of care services for elderly with disabilities in rural areas simply as lack of willingness to implement government policies(limited state), lack of care markets(absent market), and lack of NGOs and volunteers to provide care services(absent NGOs and volunteers).

If I look at welfare pluralism that emphasize the providers of services from an economic perspective, I can think of them as being strictly based on their demand. From this point of view, the lack of care services for the elderly with disabilities in rural areas can be understood as a lack of demand in the areas under study rather than a limited state, absent market, and absent NGOs and volunteers.

Therefore, in order for the researchers' argument to be more convincing, it is necessary to present the current status of the elderly people with disabilities population of study area(Jinan) in the research design section (114-116) to show that the elderly with disabilities have sufficient demand for care services.

2.

Table 2. Description of research 'participants' -> questions?

Author Response

Thank you very much for your careful review.

First, we added some data of study area to certify that the elderly with disabilities have sufficient demand for care services (127-133).  

Second, the right expression is “Semi-structured interview schedule” (183). I’m sorry that I was too careless and made a mistake when copying the article into the template.

Again, we would like to express our heartfelt thanks to the reviewer as the comments and suggestions really helped us revise and polish our paper.

Reviewer 2 Report

Authors wrote a thoroughly articulated introduction using theories. I recommended to rephrase it in the last paragraph of the introduction, or add a short paragraph why you want to study it stating the objective and the rationale concisely.

I found the method and results sessions are well presented. Although authors said they applied thematic analysis, I found that narrative analysis was also used.

Please give a clue that how you decided to stop interviewing, sensing that data has been saturated.

Conclusion:

Please rewrite conclusion. Some sentences are too long. example the first sentence of the conclusion 

"This research, based on a qualitative method in China, started with a welfare pluralism approach conducting a systematic analytic investigation into multi-dimensional accessibility barriers and the reasons behind them in care services for rural elderly people with disabilities. "

Your inference under the subtitle conclusion is  good. However, to put all of those in a paragraph is too condense. I think the two paragraphs in the conclusion can be continuation of discussion. 

Strength and limitation were reported. That can be part of the discussion.

It is obvious that authors have to reflect what should be the conclusion again. Then I recommend to write a new conclusion.

Please make it conclusive. 

Thank you very much for the opportunity to review your interesting work.

Author Response

We are extremely grateful for the insightful comments and useful suggestions. We have taken all comments very seriously and put much effort into making the following revisions (all of the revised parts are highlighted in red in the manuscript).

  1. In order to more clearly express our research purpose and research approach, we added one paragraph to explain the reason why we studied it and rephrased the last paragraph of the introduction. (80-110)
  2. In response to the comment about data analysis method, we added more information to explain why we choosethematic analysis method. (187-192)
  3. In response to the comment about data saturation, we added more details to explain at what point we stopped interviewing (151-155).
  4. In response to the comment about conclusions, we wrote a new conclusion. Also, we put the limitations in the discussion section to make the conclusion section more consistent and conclusive. (467-497)

Again, we would like to express our heartfelt thanks to the reviewer as the comments and suggestions really helped us revise and polish our paper.

Reviewer 3 Report

This is a really nice paper. It takes an important and invisible topic, such as care for older people with disabilities in rural settings. It is very well-written and I enjoyed the reading. Besides some minor comments (clarifications) in the text, my general comments are:

  • Authors need to justify some methodological choices, regarding both the data collection and data analysis strategies. Why these choices (and not the alternatives) are relevant to answer the research question?
  • Authors need to clarify whether they got IRB approval to carry out the study. Several standard procedures are described, but no reference to the approval by an ethics committee.
  • Finally, although the study looks at a very specific population (two rural villages in China), I think it has important implications for other settings both in China and in other countries. I think a discussion on the potential external validity or the relevance of the results in other settings/ countries should improve the article. 

Author Response

We are extremely grateful for the insightful comments and useful suggestions. We have taken all comments very seriously and put much effort into making the following revisions (all of the revised parts are highlighted in red in the manuscript).

  1. In response to the comment about methodology, we added more information to explain why we choose in-depth interviews (121-125) and qualitative thematic analysis method (187-192) in this study.
  2. In order to be clearer, we changed “the interviewers” to “The first author and second author” in line 117.
  3.    There is no Institutional Review Board (IRB) for ethical statement, because only medical experiments require it, according to the regulations in my university. But we provided a scanned copy of the Proof Document of Exemption to IJERPH Editorial Office.
  4. We agree with the suggestionthat it is important to refer to the results on settings both in China and in other countries. Therefore, we added one paragraph to discuss the potential external validity (393-401). But we must admit that this requires more international comparative research, which we cannot discuss deeply in this article. This is what we will do in our next research.

Again, we would like to express our heartfelt thanks to the reviewer as the comments and suggestions really helped us revise and polish our paper.
